# Dissecting Multiple Pathways in the Relaxation Dynamics of Helix <==> Coil Transitions with Optimum Dimensionality Reduction

**DOI:** 10.3390/biom11091351

**Published:** 2021-09-12

**Authors:** Gouri S. Jas, Ed W. Childs, C. Russell Middaugh, Krzysztof Kuczera

**Affiliations:** 1Department of Pharmaceutical Chemistry, The University of Kansas, Lawrence, KS 66047, USA; middaugh@ku.edu; 2Department of Surgery, Morehouse School of Medicine, Atlanta, GA 30310, USA; echilds@msm.edu; 3Department of Chemistry, The University of Kansas, Lawrence, KS 66045, USA; kkuczera@ku.edu; 4Department of Molecular Biosciences, The University of Kansas, Lawrence, KS 66045, USA

**Keywords:** laser temperature jump, molecular dynamics, dimensionality reduction, kinetics

## Abstract

Fast kinetic experiments with dramatically improved time resolution have contributed significantly to understanding the fundamental processes in protein folding pathways involving the formation of a-helices and b-hairpin, contact formation, and overall collapse of the peptide chain. Interpretation of experimental results through application of a simple statistical mechanical model was key to this understanding. Atomistic description of all events observed in the experimental findings was challenging. Recent advancements in theory, more sophisticated algorithms, and a true long-term trajectory made way for an atomically detailed description of kinetics, examining folding pathways, validating experimental results, and reporting new findings for a wide range of molecular processes in biophysical chemistry. This review describes how optimum dimensionality reduction theory can construct a simplified coarse-grained model with low dimensionality involving a kinetic matrix that captures novel insights into folding pathways. A set of metastable states derived from molecular dynamics analysis generate an optimally reduced dimensionality rate matrix following transition pathway analysis. Analysis of the actual long-term simulation trajectory extracts a relaxation time directly comparable to the experimental results and confirms the validity of the combined approach. The application of the theory is discussed and illustrated using several examples of helix <==> coil transition pathways. This paper focuses primarily on a combined approach of time-resolved experiments and long-term molecular dynamics simulation from our ongoing work.

## 1. Introduction

Formation of the functional form of a protein from initiation of a primary sequence and propagation through secondary structural elements is one of the most fundamental biochemical processes in biological systems. Understanding the mechanism of protein structure formation from its amino acid sequence [1,2,3,4,5,6,7,8,9,10,11,12,13,14,15] and predicting the three-dimensional structure from a given sequence [16,17,18,19,20,21,22,23,24,25,26,27,28,29,30,31,32,33,34,35,36] were two critical questions that attracted scientists from a variety of scientific fields. The quest for a solution to these intellectually challenging questions encouraged an interdisciplinary approach. Significant advances with new experimental approaches and the development of sophisticated theory have provided a much more detailed picture involving the complex pathway of structure formation through folding and the ability to predict three-dimensional structure [37,38]. Experimental studies to characterize the kinetics and thermodynamics of folding in the simplest systems have been very important in establishing a baseline for investigating more complex systems [38,39,40,41]. Experiments with dramatically improved time resolution have facilitated the use of fast kinetic measurements to study the relaxation dynamics of secondary structural elements and single-domain proteins in the sub-nanosecond to millisecond timescale [17,42,43,44,45,46]. These fast kinetic measurements have also been critical in providing a benchmark for a reality check of simulation studies. Analysis of all-atom molecular dynamics simulation results to study relaxation dynamics validated at an experimentally measured very short timescale can potentially provide an atomically detailed description of actual measured kinetic data [47,48,49,50]. Atomic-level descriptions strongly influence the microscopic picture of folding dynamics from molecular dynamics simulation results. It is essential to have measured timescales that can be readily compared with simulations. The advancement of theoretical studies with the energy landscape approach to protein folding was another driving force behind the development of fast-folding kinetic measurements [51,52,53,54,55,56]. Results from free energy surface calculations suggested that the barriers to protein folding should be quite small, and for the fastest folding proteins, the barrier may disappear altogether. This barrierless process may produce non-exponential kinetics, suggesting a non-two-state process during folding events. The relaxation dynamics of secondary structural elements, such as helix, hairpin, turn, and the associated timescale, can essentially capture the global structure formation pathway of a protein [44,57,58].

Helices are the most common secondary structural elements in many functional proteins and have attracted much interest when examining the formation kinetics from a coil structure. Measurements describing the timescale to a helix from a coiled structure have been substantially improved with the availability of spectrophotometers with a dramatically improved time resolution. Early studies showed that helix–coil transitions occur in the 100-nanosecond range with a model helix near room temperature. Ultrafast laser temperature jump spectroscopy has been used to measure the relaxation dynamics of helix formation in 5- to 21-residue and longer helical peptides with picosecond to microsecond resolution [42,43,44,45,46]. These newly measured kinetics with much faster time resolution have been able to identify new details in the helix formation pathway, such as non-exponential kinetic processes [50]. One of the more interesting experimental findings has been the detection of a stable helix formation in alanine-based homopolymers in solution at physiological pH [59]. An additional intriguing finding was obtained through measurement of the relaxation dynamics, previously unavailable, of a five-residue helix forming a heteropeptide with ultrafast laser t-jump, providing an important benchmark that can be easily compared with simulation times [45,46].

An extensive explanation of the approaches to modeling the helix–coil transition was described earlier [47]. A description of this biomolecular phenomenon involving statistical models [60] and the transition path was investigated earlier [61]. A true long-term molecular dynamics simulation can provide an atomically detailed description of kinetics events by extracting the rate of transitions and the nature of microscopic pathways and intermediates and can provide additional input to our theoretical understanding. Recent advancements in theory, more sophisticated algorithms, and a long-term trajectory made possible an atomistic description of kinetics and pathways and validation of experimental results, as well as describing new findings for a wide range of molecular processes [62,63,64,65,66,67,68,69,70,71,72,73].

Even as the state-of-the-art experimental techniques achieve faster time resolution, advanced algorithms and theoretical approaches are reaching a new limit, including milestoning [64,66,67,70,71,72] and Markov state modeling (MSM) [74], and microscopically detailed characterization of helix–coil kinetics and the formation mechanism is still challenging. A recent novel approach to kinetic coarse-graining with the Perron Cluster Cluster Analysis [75] (PCCA)^+^, often used in MSM, and the development of optimal dimensionality reduction (ODR) have been particularly insightful. ODR [76] was applied to a five-residue alanine-based peptide to describe folding events considering 2–4 metastable states comparable to a complex master equation model grounded on atomistic molecular dynamics simulations [77]. The contribution of the combined studies with experiments and long-term molecular dynamics simulation and the application of ODR to our understanding of helix–coil transition through multiple pathways is the subject of this review. 

Although these novel approaches have revealed previously unobserved processes in the field of protein folding, they have also recently uncovered significant new insights into folding mechanisms. These findings encompass a microscopic understanding of the folding pathways of secondary structure formation and the introduction of a measured true upper limit on the rate of protein folding (a “speed limit”), continuously downhill fast-folding kinetics, and direct observation of polypeptide collapse. One of the critical findings of this ODR modeling result is that both the helix and coil exist in a heterogeneous state, and it achieved structural characterization of multiple metastable folding intermediates. ODR also identified that helix folding occurs through multiple pathways, with the number of paths increasing with the model’s increased resolution.

## 2. Helix–Coil Kinetics

Helix–coil transitions have been studied experimentally and theoretically for nearly 60 years. In the late 1980s, many peptides comprising similar lengths and sequences found in proteins seemed to form stable helices in an aqueous environment. Since then, a large volume of equilibrium experiments and theoretical studies have been devoted to investigating the formation mechanism of a-helices in the steady state. Alanine-based peptides formed the backbone of these studies as they have a higher propensity to form helices. Kinetic studies, primarily with a 21-residue alanine-based peptide with an attached fluorescent probe at the N-terminus (X-(A)_5_ (A_3_RA)_3_-A-NH_2_), were performed with a nanosecond laser temperature jump apparatus [43]. The result was inclusive with the presence of an unexplained observed kinetic component. This observation prompted further experiments to understand helix–coil transition kinetics with an improved a-helical heteropeptide Ac-WA_3_H(A_3_RA)_3_-A-NH_2_, where the first turn of helix formation was monitored via quenching of tryptophan (W) fluorescence by a protonated histidine (H) residue four residues away [44]. Laser t-jump in the nanosecond to microsecond timescale was employed to measure structure formation rates and extract insights into the helix–coil transition pathway after initiation of the temperature jump. At the final temperatures after the jump, measured kinetic traces showed distinct non-exponential kinetics with a fast component clearly above the instrument time resolution of six nanoseconds (Figure 1A–C). Comparison of a single and a bi-exponential fit to the measured data showed that the bi-exponential fit (orange) could completely describe the kinetics instead of the single exponential fit (green). Analysis of the log of rates vs. 1/T showed a nonlinear process (Figure 1D), indicating a deviation from a two-state kinetic process. These experimental observations established the groundwork for the possible presence of intermediates along the transition pathway with the presence of faster kinetic components in this α-helical system. They also prompted an important question of whether these species, when transitioning from coil to helix through the formation of intermediates, lie in a homogeneous or heterogeneous state, pointing to the possible existence of multiple pathways along the transition pathways.

## 3. Multiple Pathways

Over many decades, the fundamental question in the protein folding field has been that of whether secondary and tertiary structures of proteins fold following a predetermined sequential step or through multiple alternate pathways on a downhill free energy surface. Experimental observations, molecular dynamics simulations, and theoretical frameworks suggest that proteins fold by multiple pathways. 

It has been computationally demonstrated that at an atomistic level, no two trajectories are identical when progressing from an unfolded state to a folded state with the generation of a large number of detailed pathways to the folded state [50]. Even at a coarse-grained level, analyses of all-atom molecular dynamics simulations demonstrated that multiple pathways are present before and after the transition state for a set of small proteins. Examining the kinetics of secondary structural elements experimentally was more challenging due to the associated times being significantly faster than that for an entire protein. Laser temperature jump with improved time resolution has definitively shown intermediates’ presence, which was previously considered a simple two-state process primarily. The complexity of involving the possible pathways involved in the transition to the native helix state (Figure 2) from an unstructured coil state prevented us from visualizing multiple states due to the lack of a sophisticated theoretical framework and its application to a system where computational findings can be readily verified with experimental results. The development of the optimum dimensionality reduction theory and its application to a long-term simulation trajectory to examine the folding pathways of several significantly large α-helical homo- and heteropeptides are discussed below.

## 4. Optimum Dimensionality Reduction

Atomistic molecular dynamics simulations are employed extensively to describe events that explicate the functions in biology. A significant challenge exists in expanding their scope due to biological systems’ vast spatial and temporal landscape. In recent years, substantial efforts have been made to develop advanced theories, sophisticated algorithms, and faster hardware to address these limitations. Here, we have applied optimal dimensionality reduction (ODR) theory combined with long-term simulation to describe multistate kinetics at an atomically detailed level for the most common secondary structural elements in proteins. Time-resolved experimental results are readily available to test the validity of this approach. ODR is a systematic approach to construct a reduced description of the dynamics of aggregated superstates obtained by clustering microstates. This is commonly applicable to kinetic events with discrete states and continuous-time evolution. It further encompasses continuous space events with discrete or continuous-time evolution. This reduced model maintains an optimal balance to recover short- and long-term dynamics.

Recently, attempts have been made to describe helix–coil transitions by employing molecular dynamics simulations. The inference is made that a coil-to-helix transition occurs following multiple nucleation sites to fragmented helices. Description of helix folding rates and pathways from a pre-determined path is pursued with a milestoning approach. A complete description of helix–coil kinetics at an atomically detailed resolution was missing. Experimental results of fast kinetics are uncovering new details of the relaxation dynamics simultaneously with sophisticated theoretical approaches, which are being tested to characterize the measured pathways with Markov state modeling and milestoning. Recently developed ODR was able to describe a folding pathway, with two to four states, in a five-residue alanine homopolymer that can be compared to a master equation in a model based on molecular dynamics simulation. Interesting approaches to kinetic coarse-graining with PCCA+ are promising and open new important testing options. In our studies, a novel approach with simplified coarse-grained models of low dimensionality grounded upon a kinetic matrix was executed on the MD trajectory to capture microscopic insights into the pathway of the coil-to-helix transition. The conformations generated from analysis of the MD trajectory were clustered into a set of microstates, and a lifetime-based kinetic analysis was performed. Employing the PCCA+ algorithm [75,76,79,80], these microstates were aggregated to a small number of metastable states. These metastable states formed the foundation for an optimally reduced dimensionality rate matrix and transition path theory analysis. Low dimensionalities were modeled, considering new experimental findings into the folding pathway of an alpha-helical heteropeptide. ODR analysis showed that the helix and coil states are heterogeneous. This discovery facilitated the characterization of several intermediates in the transition pathway.

Regarding the coil-to-helix transition pathway, our analysis further indicated that helix–coil transition progresses through multiple pathways, and the number of paths increases relative to the increased resolution of the model. A brief summary of the results involving the application of this approach with new findings is presented here.

Analysis of the kinetics for a 21-residue α-helical heteropeptide in two protonation states [78,81] was based on 12–13-microsecond MD trajectories generated with GROMACS [82] involving the CHARMM36 force field [83] and TIP3P water at 300 K. Clustering of structures from the analysis of MD trajectories was performed with the gromos algorithm in GROMACS, with CA atom RMSD. A cluster radius of r_cluster_ = 4.5 Å captured N_c_ = 194 clusters for WH21^0^ and Nc = 199 clusters for WH21^+^. The lifetime-based kinetic analysis counting transitions between cluster cores and cluster residence times provided a kinetic matrix of order 194 for WH21^0^ and 199 for WH21^+^. The kinetic matrix diagonalization generated eigenvalues λ_i_ and timescales t_i_ = 1/λ_i_. The reduced kinetic matrix for the dimensionality of N = 5 presented (Figure 3) a picture of a kinetic network with a “helix” metastable set of 14 and a “coil” set of 146, and the intermediates at I1 contained 7 of the original clusters, those at I2 contained 12, and those at I3 contained 15. The most highly populated state was the helix, followed by the coils. The corresponding lifetimes were as follows: for the helix, 670 ns; coil, 720 ns; I1, 150 ns; I2, 280 ns; and I3, 230 ns. At N = 5 coarse-grained states, there were sixteen possible folding pathways, with one direct pathway, three coursing through a single intermediate, six progressing with the formation of two intermediates, and six progressing by visiting all three intermediates. 

Based on the simulation trajectory analysis, the experimentally measured fast kinetics of the helix–coil transition for the protonated WH21 (WH21^+^) relaxation time of 20 ns at 296 K were assigned to the correlated formation/breaking of neighboring hydrogen bonds. An analogous assignment was proposed for the non-protonated form of WH21 (WH21^0^). Fitting to the average hydrogen bond fluctuation autocorrelation function (Figure 4A,B) produced time constants of 14 and 310 ns in WH21^+^ and 16 and 340 ns for WH21^0^.

Hydrogen bond analysis further demonstrated a significant hydrogen bond fluctuation correlation between neighbors (Figure 5A,B). The average correlation coefficient for WH21^+^ was observed to be 0.82 with the nearest neighbors, 0.63 with the next nearest neighbors, and 0.50 for hydrogen bonds three residues apart. For WH21^0^, the average correlation coefficient was 0.84 with the nearest neighbors, 0.68 with the next nearest neighbors, and 0.57 with hydrogen bonds separated by three residues. This analysis further suggested that helical hydrogen bond breaking/formation seems to occur in groups of the 1−4 nearest neighbors. These results led us to propose that the presence of cooperative breaking/formation of small sections of the helix is a major driving force behind the helix–coil transition. These findings are consistent with those of Scheraga et al., where the folding of multi-helical structures could be described with the formation of kinks [84], and the events were more prominent than single-residue dominant folding transitions [85].

Additionally, it helped with the assignment of the observed faster signal in the helix–coil kinetics measurement from the laser t-jump experiment. Application of this procedure to an alanine-based 21-residue helical homopeptide produced similar results. The summary of these findings led us to conclude that this cooperative breaking/formation of the small segment of the helical peptide should be present along the transition pathway involving the helix, coil, and intermediates and can explain the observed specific relaxation time.

The statistical analysis of the transition pathways (Figure 6A,B) of WH21^+^ and WH21^0^ showed that in the neutral peptide (Figure 6A), the probability of helix initiation and structure formation (purple and blue) is increased at the N-terminus (NH = 0–5 region), is decreased at the C-terminus, and remains uniform in the mid-segment. In WH21^+^ (Figure 6B), the structure formation pattern differs strongly with decreased initiation and structure formation probability near the protonated histidine at the fifth residue position. The structure formation patterns in mid-segment regions (NH = 8–12) are markedly different from the protonated form. Analysis of WH21^+^ showed that the helix–coil transition progresses through an off-pathway intermediate with hydrogen bonds at residues 11–16 and favors N-terminal helix formation. In the case of WH21^0^, our analysis showed a higher propensity for helix formation near the central segment of the peptide and frayed at both ends.

Application of optimum dimensionality reduction (ODR) to a length-dependent helical homopeptide [86] provided an opportunity for direct microscopic examination of helix–coil transitions relative to the changes in amino acid composition. Analysis of the several microsecond long trajectories showed a gradual increase in helicity from A_5_ to A_21_. Detailed analysis of the rates for the coil-to-helix and helix-to-coil transitions showed that the rates decrease with increased chain length (Figure 7A). Hydrogen bond population analysis revealed that the central segment of these homopeptides has the highest propensity to form a helix, as observed in the helical heteropeptides (Figure 7B,C). Coarse-grained kinetics employing ODR showed primarily a two-state process in the helix–coil transition in A5 and A8. Even in these short peptides, the 3_10_-helix population was an important finding in the folding path [86].

In contrast, A15 and A21 progress through intermediates well described in the transition pathways (Figure 7C,D). Similar to our observation in helical heteropeptides, helical homopeptides follow multiple pathways as they transition from coil to helix, as is evident from the presence of the inhomogeneous-state helix, coil, and intermediate. This heterogeneity is present in the two-state and higher dimension models that involved helix–coil transitions. Analysis of statistics for the coil-to-helix transition from MD trajectories showed the formation of a core helical structure with three hydrogen bonds uniformly across the chain, with the most stable form of helix appearing to exist in the central segment of the chain, which propagates towards the termini.

The longer alanine homopeptide intermediates primarily include partially folded helices at termini and turns, with a small population of partial hairpins. There is also a small probability of sampling 3_10_-helix structures in these systems, typically in the three-residue nuclei. π-Helices were not found in our trajectories.

Generally, we refer to the model systems under study here as “helix-forming peptides”. It is clear from both experimental and computational studies that these small peptides do not form unique helical structures. Instead, they exist in dynamic equilibria of multiple conformations with a dominant helical population. For instance, at the melting temperature of 297 K, WH21^+^ exists with a 50% population of a-helical conformations [81]. It is worth mentioning that our computational measure of helix fraction, similarly to circular dichroism or infrared spectroscopic determination, determined the fraction of residues (or hydrogen bonds) in the helical conformation. For MD trajectories with ca. 50% helix fraction calculated in this way, the population of the fully helical form is much lower, typically 5% or less. Thus, a majority of the helical forms sampled are partially formed. This finding is consistent with the heterogeneous “helix” states determined with ODR.

It is important to examine the place of optimal dimensionality reduction (ODR) to milestoning and Markov state modeling (MSM) for an atomistic understanding of kinetics. MSM and milestoning are grounded on a more mature theoretical foundation as they have been present for longer. Milestoning can be used to describe processes with arbitrarily long timescales requiring large numbers of short trajectories between milestones, with the significant disadvantage of the technical difficulty of the method which has made it more challenging to automate. Both MSM and ODR explore conformational space with a smaller number of relatively long trajectories and apply only to molecular systems of moderate size. The advantage of MSM is the availability of relatively easy-to-use tools. Structural interpretation of MSM results is still a challenge. The use a relatively small number of microstates to discretize the conformational space and the ease of structural interpretation make our approach of ODR attractive. The current challenge with ODR is the incomplete exploration of the mathematical and physical properties of the method. Importantly, our investigation employing ODR uncovered the heterogeneity of the helical state and characterized novel folding intermediates on the coil-to-helix pathway. It should be noted that an essential advantage of ODR is that it generates low-dimensional kinetic models which reproduce the slowest timescales of the system dynamics, as well as generating a free energy landscape. Alternate approaches with reduced dimensionality are oriented explicitly to the landscape [87,88,89].

ODR reproduces the slowest timescales from MD trajectories significantly well, and a good agreement is present between calculated and experimental folding times for small–medium-sized peptides with the application of modern all-atom protein potentials. For instance, for WH210 at 300 K, relaxation times of about 300 ns were obtained with OPLS/AA [51], AMBER [51], and CHARMM [78] potentials, close to the observed result of 280 ns at 296 K [78]. In the case of the WH5 pentapeptide, the computed relaxation time of 4–5 ns [90] correlated well with measured values of ca. 10 ns [45,46].

## 5. Summary

Atomically detailed kinetics simulations are important descriptive tools for time-dependent events in biophysical chemistry, as life processes occur as a function of time. Fast kinetic experiments have significantly facilitated our understanding of the fundamental processes in folding pathways involving α-helices, β-hairpin, contact formation, and overall collapse of the peptide chain. The atomistic description of the observed events has been inadequate. The recent availability of advanced theories, robust algorithms, and true long-term trajectories will uncover microscopically detailed kinetics processes by examining folding pathways and through validation of experimental results with exciting new findings for a vast array of processes in molecular biophysics. We have discussed the successful application of optimum dimensionality reduction theory and the resultant new findings. We are taking advantage of this newly developed approach by progressively increasing the complexity of biomolecular systems from a simple dipeptide to small single-domain proteins through secondary structural elements. These approaches should gradually make possible the investigation of more complex cellular processes. 

## Figures and Tables

**Figure 1 biomolecules-11-01351-f001:**
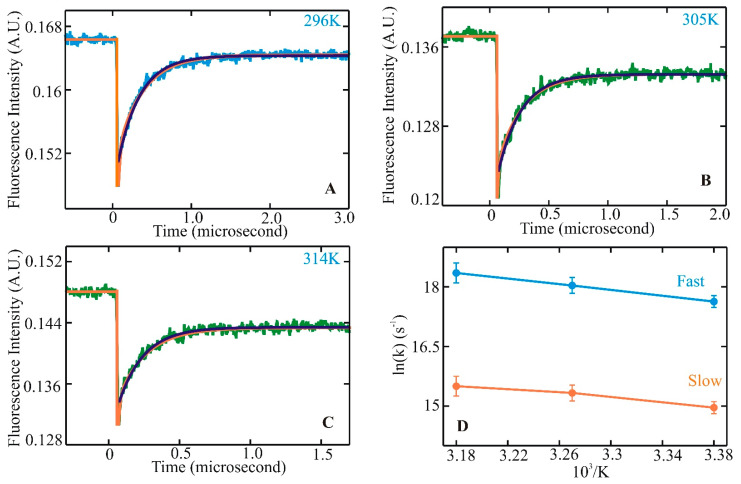
Helix–coil kinetics measured for WH21^0^ following laser t-jumps. Kinetic traces following relaxation dynamics at final temperatures of 296 (**A**), 305 (**B**), and 314 K (**C**). Tryptophan fluorescence intensity change is plotted against the change in time. The single exponential fit to the data is shown in green. The bi-exponential fit to the measured kinetic trace is represented in orange for WH21 at 296, 305, and 314 K. The WH21 concentration in the measured sample is ~100 μM. Kinetic rates in Arrhenius form, ln(k) vs. 1/T, are shown in (**D**), where the solid blue line represents the fast component, and the slower component is in orange [78].

**Figure 2 biomolecules-11-01351-f002:**
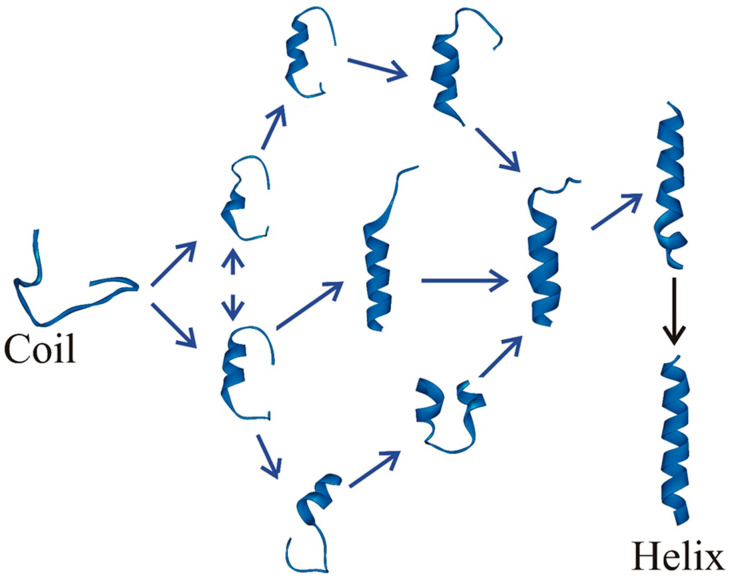
Schematic representation showing the progression pathways from the coil state to the helix state during helix–coil transition. The coil state is racing to reach the native helix state by assuming multiple conformations and coursing through various pathways. The coil state can become the helix state along the on-pathway through the formation of intermediates. It also has a high probability of reaching a helical conformation by following the off-pathway and visiting intermediates.

**Figure 3 biomolecules-11-01351-f003:**
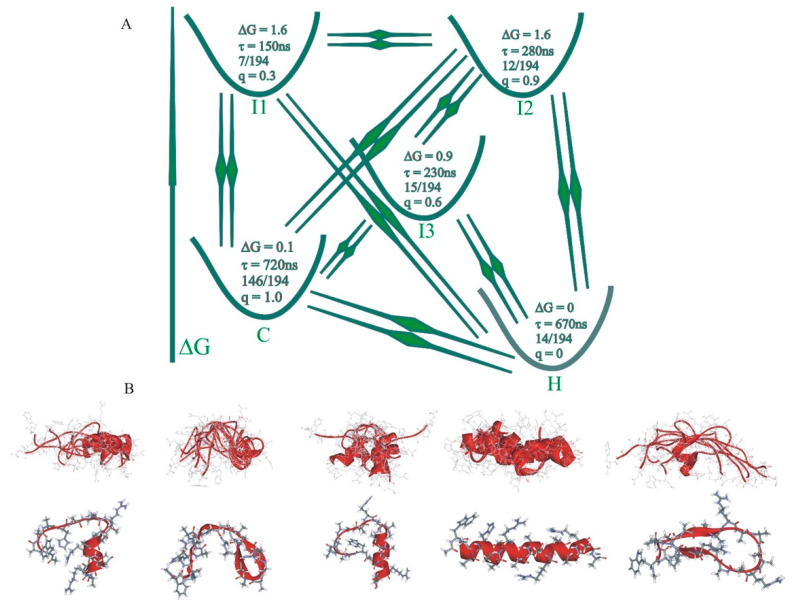
Kinetic network diagram. Analysis of the reduced dimensionality model for WH21^0^ with N = 5 coarse-grained states. (**A**) The kinetic connectivity representing the R matrix. (**B**) The structures from the corresponding cluster center. In B, the top row represents a complete ensemble of cluster centers for each metastable state. The row at the bottom shows the central structure of the most highly populated cluster in each metastable state [78].

**Figure 4 biomolecules-11-01351-f004:**
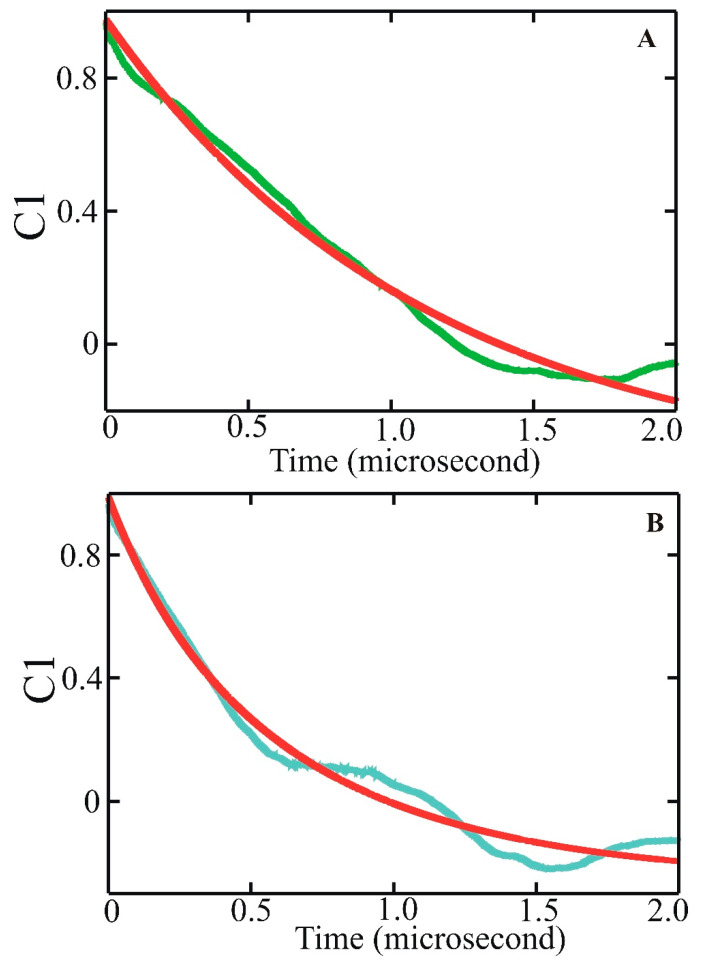
The average autocorrelation functions (ACFs) for length fluctuations of the 19 helical hydrogen bonds (C==Oi · · · H–Ni+4). (**A**) For WH21^0^: average autocorrelation in green with bi-exponential fit in red. (**B**) For WH21+: average autocorrelation in cyan with bi-exponential fit in red [81].

**Figure 5 biomolecules-11-01351-f005:**
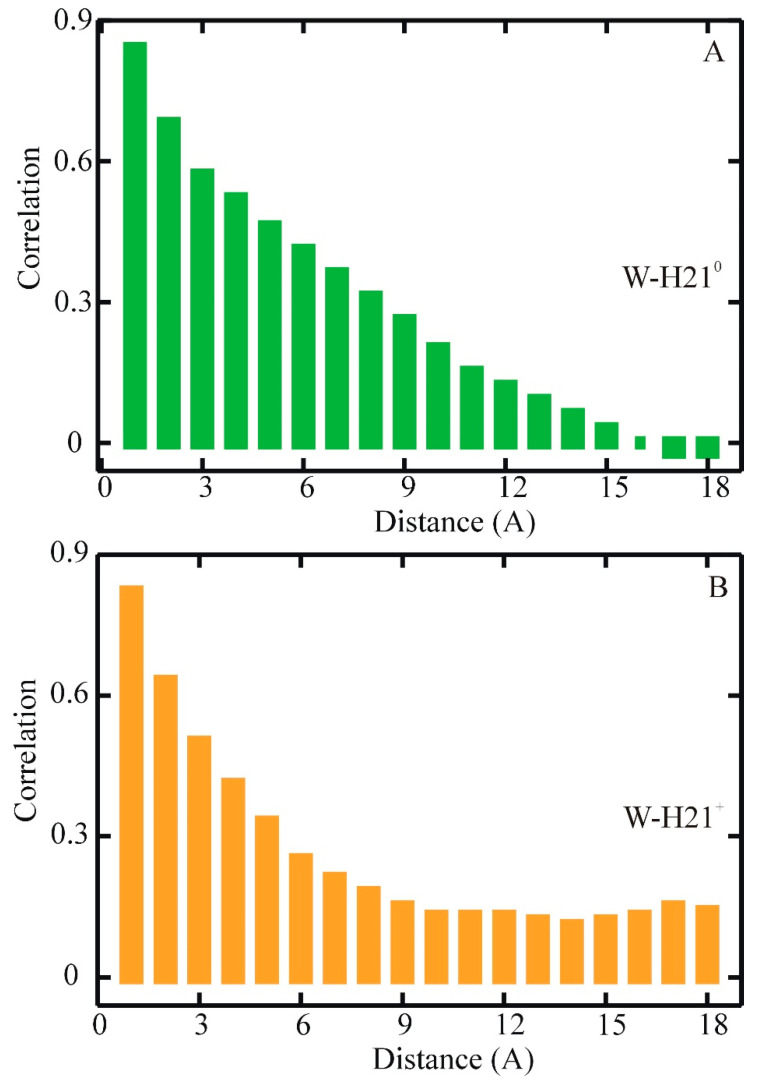
Correlation coefficients for intrachain fluctuations of hydrogen bonds as a function of the neighboring separation distance in sequence space. The displayed results are from the averaged data from hydrogen bonds, with the starting residues i differing by 1, 2, 3, …, 18 residues. (**A**) represents WH21^0^, and (**B**) is WH21^+^ [81].

**Figure 6 biomolecules-11-01351-f006:**
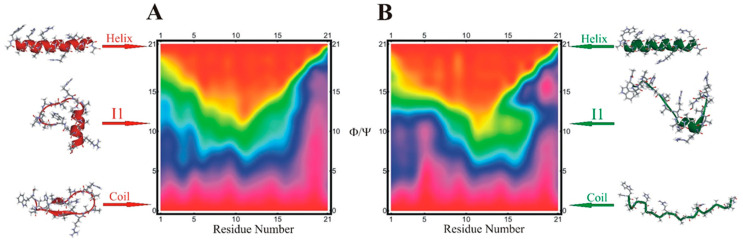
Results of the statistical pathway analysis for the helix–coil transition are shown. Data for the helix populations of individual residues (horizontal axis, NR = 1–21) are averaged over samples of conformations with a given number of residues in the helical region of the Ramachandran plot (vertical axis, NH = 0–21)—i.e., within a 20^o^ radius of the ideal helix location taken from the final 500 ns of all 40 replicas of replica exchange molecular dynamics. Thus, the bottom row (labeled “0” on the vertical axis) describes structures with NH = 0 helical residues. The top row, labeled “21”, describes structures with all 21 residues in the helical state. The colors encode the average helix population of each peptide residue in the 0–1 range; red—0.0, green—0.5, purple—1.0. The molecular structures are cluster centers for the N = 3 model. (**A**) WH21^0^ and (**B**) protonated WH21^+^ [81].

**Figure 7 biomolecules-11-01351-f007:**
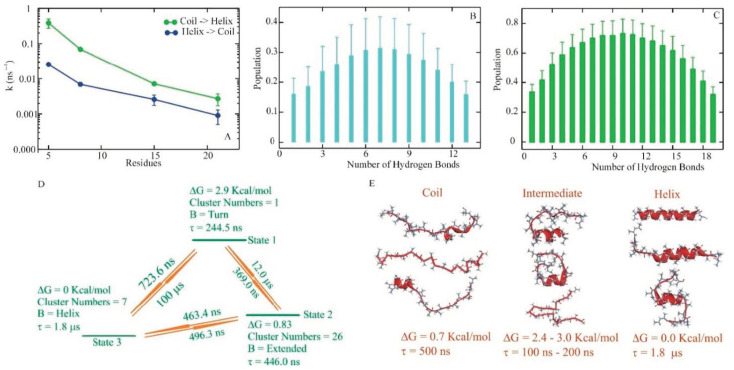
(**A**) Helix–coil kinetics from molecular dynamics simulation. Rates for coil-> helix k_f_ (green) and helix->coil k_u_ (blue) for A5, A8, A15, and A21. (**B**,**C**) Average helical hydrogen bond populations for A15 and A21. A hydrogen bond is defined as an interaction between the C=O of residue i and NH of residue i + 4. Error analysis provided 95% confidence intervals. (**D**) An example of a coarse-grained kinetic model with N = 3 dimensions for A21, based on N_c_ = 34 initial clusters. (**E**) A summary of the helix, coil, and intermediate structural properties from the aggregated states in ODR models for A21 at various resolutions [86].

## Data Availability

Data presented in this study are available on request from the corresponding author. The data are not publicly available due to large volume.

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
