# Peer review of "Dissecting Multiple Pathways in the Relaxation Dynamics of Helix <==> Coil Transitions with Optimum Dimensionality Reduction"

_biomolecules, 2021, doi:10.3390/biom11091351_

Round 1
Reviewer 1 Report
Overall remarks:
The manuscript represents a review focusing on a crucial structural phenomenon in biology – helix to coil transition. It includes the published theoretical studies, methodological developments as well as empirical investigations. As a computing methodology, the Authors pointed principally to the Optimum Dimensionality Reduction technique. First, as this technique is not widely used among the data dimensionality reduction techniques, it should be introduced better, in particular, the choice of this technique or preference should be clearly explained. Second, it is surprising that a very detailed representation in the manuscript concerns only the Authors proper papers, while the really fundamental research works in the domain (helix to coil transition) are only briefly mentioned. Even if the manuscript is oriented on the personal results, it should be equilibrated by presentation of the results obtained by the other scientists with using different linear and non-linear dimensionality reduction techniques (e.g., https://doi.org/10.3389/fmolb.2019.00046; DOI:10.1007/978-3-030-17935-9_21 ; doi: 10.1002/prot.22526).
Major remarks:
- A single sentence that summered 1-36 references in Introduction is so short and should be considerably improved by differentiation of these references on the distinct topics.
- The manuscript considers only the α-helix to coil transition, while in Fig. 3 one conformation presents clearly a short antiparallel sheet formed by two β-strands. In this case, an analysis of the helix to strand transition will be useful.
- Similarly, the helical transformation to coil (and vice versa) is accompanied by formation of the transient structures - 310 and π-helices – at least on the helix extremities, the effect not considered in the manuscript.
- The multiple folding pathways and complex kinetics may be related to the intrinsic disorder, also do not mentioned in the manuscript.
Minor remarks:
- The figure captions should be completed with references to original articles from which the related figures are reprinted or adapted.
- Figure 1A-D (p. 4): Why is the measured data line for T = 296 K in blue, while for others it is green? Would it be more consistent to use a normalized fluorescence intensity unit for all shown temperatures? The unit of the x-axis is [1000/K] rather than [K].
- Line 128: According to your own studies (in the references), the bi-exponential fitting of the fluorescent decay rates could also indicate the deviation from the two-state kinetic process and it would be more pronounced also to mention here that fact in support of the non-two state model.
- Line 218: It is stated that the analysis is based on “several 20 µs MD trajectory” (trajectories?) but in the mentioned references (78, 79) the 12 and 13 µs long MD simulations were reported. Do the Authors present in the manuscript the new (updated) results and their conclusions are based on the extended 20 µs simulations for two forms of the heteropeptide compared to references 78, 79?
- The annotation of the helices and harpins will be better in Greek (α- and β).
- Figure 3. The superimposed conformations for each metastable state should be distinguished by different colours.
- Figure 3, 4, 6: In the captions the superscript symbols are not properly typesetted for both non-protonated and protonated forms of WH21.
- Figure 5: Please consider changing the x-axis label of the figure to a more explicit one, for example, “Distance to neighbour'' as in Ref. 79. Also, the last three bars in the subfigure A are not correctly plotted, the values are started not from 0.
- Line 266. Results about the termini residues are artefacts from your modelling and simulations should be handled with care. The following results about the better propensity at helices formation in the middle of the peptide is rather intuitive as the termini are generally in coiled structures
- Figure 7 B-C : the x and y-scales are misleading. You should put the same scale for both graphs to better illustrate your interpretation, line 289.
- Typos : Figure 6 legend : 20o radius, maybe 20o radius ? WH210 instead of WH210
Author Response
Response to Reviewer 1:
General remarks:
Our manuscript focuses primarily on our recent results for describing the microscopic details of helix-coil transitions with the new Optimal Dimensionality Reduction (ODR) approach. Due to the large body of both experimental and computational results on this topic, writing a fully comprehensive review would not easily fit into the size of a typical scientific article. At the end of the manuscript, we have included a section comparing ODR to other methods describing kinetics and free energy landscapes – Markov State Modeling and Milestoning. It is worth underscoring that ODR produces low-dimensional models with the optimal reproduction of the system’s slowest dynamical timescales. The articles that the referee brings to our attention deal only with the description of the landscapes, and do not describe kinetics, so they are not directly comparable to our approach. We have added references to these works at the end of the Helix-Coil section as examples of alternative dimensionality reduction approaches while also stressing the unique properties of the ODR method.
Major remarks:
- We have specified the sentence with specific references corresponding to the research focus in the sentence rather than putting the references at the end of the sentence for the purpose of this review (1-36).
- There are rare beta structure occurrences in the longer peptides' trajectories (WH21, A21, and A15), and we do not have sufficient statistics to build kinetic models involving specifically beta-strand to helix transitions. Typical representatives are two isolated beta-bridges or two pairs of beta-sheet residues, with an even rarer occurrence of up to six residues in beta-sheet form. All of these states make 0.1% or less of the sampled structures. Unstructured turns are much more highly populated intermediates in the folding pathway. A brief discussion of beta structure has been added in the section describing folding intermediates.
- n our data, we do find a small presence of 310 helices. These are typically various three-residue helical nuclei located along the whole length of the peptide chains. The total populations of these 310 helix states are about 1%. A central turn with helical character is an important structure on the folding landscape of shorter peptides, like A5 and A8 (see our Life, 2021 article for more details). Similarly, Huo and Straub, 1999 (a new reference added as part of this review) did not find a major 310 contribution to A10 folding. There is no evidence for pi helices in our simulations. The preference for pi helices was a property of pre-2000 protein force fields and is not much seen in the latest parametrizations. A comment about 310 and pi structures was added in the discussion of folding intermediates.
- The referee is correct, but this may be a question of nomenclature. Intrinsically disordered proteins (IDPs) are proteins that do not spontaneously fold into well-defined 3D structures. The small model peptides that are the objects of our studies typically exhibit fractional helix populations of 40-60% (the longer ones - WH21, A21) or even 5-12% (shorter ones - A5, A8). Thus, all of these systems may be considered in terms of the dominant population since they do not assume a single stable structure. We tend to attach the more optimistic label of “helix forming” to these peptides since they occasionally form helices and may be used as experimental and computational models for helix-coil transitions. It is worth noting that the helix fraction measures the average number of residues in the helical conformation; this is true for both simulations and measurements like CD or IR. Our MD data indicate that in trajectories with helix fractions of close to 50%, the fully formed alpha helix population is much lower, typically 5% or less. Thus, most of the observed/computed helix fraction appears to come from partially folded intermediates. This effect is seen in our ODR kinetic modeling, where the “Helix” coarse-grained aggregate states are heterogeneous, combining several partly helical conformations. We have added a brief discussion of this topic to the end of the Helix-Coil section.
Minor comments
- References to original publications have been added to figures.
- We have improved Figure 1 according to the reviewer’s suggestions. The y-axis scale is chosen to best show the variation of the data.
- We have added a comment about non-two-state-kinetics to the text.
- The reviewer is correct, and the trajectory lengths were 12-13 . This typographical error has been corrected.
- We have changed most ‘alpha’ and ‘beta’ occurrences to Greek letters.
- We have added labels to the structures in Figure 3. Showing ensemble members in different colors is generally a good idea. However, in our case of a large number of ensemble members and a small size of the image, we thought the single-color images might be easier to interpret by viewers.
- The superscripts have been corrected.
- We have changed the label for Figure 5. The last three bars actually show correlation coefficients that are close to zero (the values are 0.00, -0.02, and -0.02).
- These are good points. It is not easy to see in the heatmaps of Fig. 6, but the raw data for most of our studied peptides show a slight preference for the formation of the first helical hydrogen bond at either the N- or C-terminus relative to the rest of the chain. The further discussion is meant to underline the difference between the protonated and neutral forms of WH21. Namely, the strong His…Trp interactions in the protonated form stabilize hydrogen bonding at the N-terminus and move the stable helix nucleus to an off-center position. The discussion has been slightly modified to reflect this.
- It is a trade-off between consistency and showing the range variation in the data. We believe using the same scales in Fig. 7B-C would make 7B hard to read. The difference in scales has been noted in the figure caption.
- The typos have been corrected.
Reviewer 2 Report
This paper briefly reviews some experimental and simulation methods currently employed to study protein folding dynamics/kinetics and reproduces recent results obtained by the authors for the kinetics of coil-helix transition in a helical heteropeptide (WH21). The authors start with the experimental study of kinetics of WH21 after laser T-jump, which indicated the presence of intermediates in the process of helix-coil transition and stimulated subsequent simulation study of this process. Then, they describe the results of the MD simulation of coil-helix transition in neutral and protonated WH21 peptides, construct coarse-grained low-dimensional kinetic networks based on the simulated MD trajectories, and analyze kinetics of coil-helix transition using these networks.
General comments:
- It would be useful to see what has been known about coil-helix transition pathways from other works (e.g., Huo and Straub, Proteins 36, 1999, 249).
- Since all figures are simple copies of figures from published articles [78,79,82], it would be appropriate to indicate in the captions to these figures the articles from which they are taken. This will allow the potential reader to read a more detailed discussion of these figures, if necessary.
Other comments:
- Lines 26-27: According to the statement that “Analysis of the actual long-time simulation trajectory extracts a relaxation time directly comparable to the experimental results”, it would be interesting to have some examples of this.
- Lines78-84: It seems evident that among the “true long-time molecular dynamics simulations” the works of D. E. Shaw and coworkers should be mentioned.
- Line 208 (Fig.7): The representative peptide structures (two bottom lines) are not labeled.
- Lines 218-219: “Analysis of the kinetics for a 21-residues alpha-helical heteropeptide in two protonation states [78,79] is based on several 20 μs MD trajectory …”. At the same time, according to Refs. [78,79], the trajectory lengths were 12-13 μs. What is correct?
- Line 256 (Fig. 5): What is the correlation coefficient for the fluctuations of hydrogen bonds?
- Lines 292-293: “Coarse-grained kinetics employing ODR showed a two-state transition in the helix−coil transition in A5 and A8 primarily.” A reference to this statement?
Author Response
Response to Reviewer 2:
- A brief section describing some alternative approaches to modeling peptide folding has been added to the Introduction, including the mentioned work of Huo and Straub.
- We have added citations to the figures adapted from published work.
- We have added a brief section describing agreement of simulated and measured peptide folding times to the Conclusions.
- We have added a citation to a review paper from the D.E. Shaw group to reflect their important contributions to long-term molecular dynamics simulations.
- We believe the reviewer refers to Fig. 3, which indeed has missing structure labels. Labels of ‘I1’, ‘I2’, ‘I3’, ‘Helix’ and ‘Coil’ have been added in panel 3B.
- The reviewer is correct, the trajectory lengths were 12-13 . This has been corrected in the text.
- The correlation coefficient is just a simple Pearson correlation between the fluctuation data sets for two hydrogen bonds. For completeness, the equation used is
Here denotes the length of hydrogen bond i at time tk, and <…> denotes a trajectory average.
- A reference to our Life 2021 article has been added.
Reviewer 3 Report
This is a very interesting paper which addresses the mechanism and kinetics of helix formation. The Authors used laser T-jump and all-atom molecular dynamics to study helix formation in two host-guest peptides and MD simulations to study helix formation in poly-L-Ala peptides. They used advanced cluster analysis and dimensionality reduction to find folding intermediates and to construct the respective free-energy landscapes. The obtained results are of high interest to the community. However, there are a number of issues that must be addressed before the paper is published.
- In the Introduction (page 3 line 96), the Authors say state that the paper is a review article. Therefore, it could be expected that the source of the data displayed in Figures 1-7 should be indicated in the Figure legend. If the paper is not a review article, the statement from the Introduction should be removed and more details given about the methodology.
- I have had troubles to follow the conclusions from the helix fraction maps of Figure 6 (page 9, lines 266-295). If I understand the Figure legend correctly, panels A and B show the fractions of residues with indices shown in the horizontal axis under the condition that the total number of residues in helical state is that shown in the vertical axis, of the WH210 and WH21+ peptides, respectively. If so, it looks like the highest fraction of helical state is in the N- and in the C-terminus (the red region, which corresponds to zero helix fraction according to the legend) is the least expanded at the N- and at the C-terminus, while the Authors say that the probability is reduced at the C-terminus. This should be clarified. Also, including a colorscale panel (e.g., right to panel B), instead of referring to rudimentary color code in the legend, would make the Figure clearer.
- page 5, line 155. "It has been computationally demonstrated that at an atomistic level, no two trajecto-", a reference needs to be cited here.
- page 6. The sentence beginning in line 205 is incomplete.
- Figure 3 in page 6: The structures shown in panel B should be assigned to the H, C, I1, I2, and I3 states.
- page 10, line 318: "Both MSM and ODR explore the entire conformational in multi-", a noun is missing after the adjective "conformational"; perhaps the Authors meant "conformational space".
- "contract" apparently appears instead of "contact" in lines 16 of the abstract and line 330 of the text. "coliapse" instead of "collapse" appears in line 16 of the abstract.
- A related work on helix formation as a step of protein folding was carried out earlier by Niemi and coworkers by using the coarse-grained simulations interpreted in terms of dark-soliton theory (J. Chem. Phys., 2014, 140, 025101) and Maisuradze and coworkers, the analysis being performed in terms of principal component analysis (J. Chem. Theory Comput., 2013, 9, 2907-2921). The Authors might want to compared those results with theirs.
Author Response
Response to Reviewer 3:
- References to figures adapted from published work have been added. (This is analogous to Reviewer 1 comment 2.
- We have slightly expanded Fig. 6 caption to make it easier to interpret, and have added a color code bar.
- We have added a reference (Jas, Middaugh, Kuczera, J Phys Chem B 2014, 118, 639-647) that explicitly discusses differences in multiple individual helix-coil transition paths.
- The original text in lines 204-207 has been revised to correct for missing words.
- Figure 3 caption has been improved and color code added. (Analogous to Reviwer #1 comment 5).
- The original text in lines 312-318 has been revised for clarity and precision.
- The occurrences of ‘contract’ and ‘coliapse’ have been corrected.
- The references are included in a new brief section of the Introduction describing examples of alternative approaches to the helix-coil transition modeling.
Round 2
Reviewer 1 Report
The authors have responded to all my comments / remarks / questions and have improved the manuscript. This improved version can be accepted for publication.
Author Response
We have addressed the reviewer’s concern with fine spell checks.
Reviewer 3 Report
The Authors have addressed my comments satisfactorily. However, the second part of the new sentence in page 8, lines 258-260, which follows the citation of the new reference 86, is hardly comprehensible and should be fixed:
"These findings are consistent with the Scheraga et al., where folding of multi-
helical structures possibly be described with the formation of kinks86and the events are
more prominent than a single residue dominate folding transitions.87"
Author Response
We have corrected the concerned sentence “These findings are consistent with the Scheraga et al., where folding of multi-helical structures possibly be described with the formation of kinks86and the events are more prominent than a single residue dominate folding transitions.87" to make it easier to read.
Additionally, we have carried out the fine spell check, as suggested.